# Zero-Shot Recognition with Unreliable Attributes

**Dinesh Jayaraman**
University of Texas at Austin
Austin, TX 78701
dineshj@cs.utexas.edu

**Kristen Grauman**
University of Texas at Austin
Austin, TX 78701
grauman@cs.utexas.edu

## Abstract

In principle, zero-shot learning makes it possible to train a recognition model simply by specifying the category's attributes. For example, with classifiers for generic attributes like *striped* and *four-legged*, one can construct a classifier for the zebra category by enumerating which properties it possesses—even without providing zebra training images. In practice, however, the standard zero-shot paradigm suffers because attribute predictions in novel images are hard to get right. We propose a novel random forest approach to train zero-shot models that explicitly accounts for the unreliability of attribute predictions. By leveraging statistics about each attribute's error tendencies, our method obtains more robust discriminative models for the unseen classes. We further devise extensions to handle the few-shot scenario and unreliable attribute descriptions. On three datasets, we demonstrate the benefit for visual category learning with zero or few training examples, a critical domain for rare categories or categories defined on the fly.

## 1  Introduction

Visual recognition research has achieved major successes in recent years using large datasets and discriminative learning algorithms. The typical scenario assumes a multi-class task where one has ample labeled training images for each class (object, scene, etc.) of interest. However, many real-world settings do not meet these assumptions. Rather than fix the system to a closed set of thoroughly trained object detectors, one would like to acquire models for new categories with minimal effort and training examples. Doing so is essential not only to cope with the "long-tailed" distribution of objects in the world, but also to support applications where new categories emerge dynamically—for example, when a scientist defines a new phenomenon of interest to be detected in her visual data.

Zero-shot learning offers a compelling solution. In zero-shot learning, a novel class is trained via *description*—not labeled training examples [10, 18, 8]. In general, this requires the learner to have access to some mid-level semantic representation, such that a human teacher can define a novel unseen class by specifying a configuration of those semantic properties. In visual recognition, the semantic properties are *attributes* shared among categories, like *black*, *has ears*, or *rugged*. Supposing the system can predict the presence of any such attribute in novel images, then adding a new category model amounts to defining its attribute "signature" [8, 3, 18, 24, 19]. For example, even without labeling any images of zebras, one could build a zebra classifier by instructing the system that zebras are *striped*, *black and white*, etc. Interestingly, computational models for attribute-based recognition are supported by the cognitive science literature, where researchers explore how humans conceive of objects as bundles of attributes [25, 17, 5].

So, in principle, if we could perfectly predict attribute presence[1], zero-shot learning would offer an elegant solution to generating novel classifiers on the fly. The problem, however, is that we can't assume perfect attribute predictions. Visual attributes are in practice quite difficult to learn

accurately—often even more so than object categories themselves. This is because many attributes are correlated with one another (given only images of *furry brown* bears, how do we learn *furry* and *brown* separately? [6]), and abstract linguistic properties can have very diverse visual instantiations (compare a *bumpy* road to a *bumpy* rash). Thus, attribute-based zero-shot recognition remains in the "proof of concept" realm, in practice falling short of alternate transfer methods [23].

We propose an approach to train zero-shot models that explicitly accounts for the unreliability of attribute predictions. Whereas existing methods take attribute predictions at face value, our method *during training* acknowledges the known biases of the mid-level attribute models. Specifically, we develop a random forest algorithm that, given attribute signatures for each category, exploits the attribute classifiers' receiver operating characteristics to select discriminative *and* predictable decision nodes. We further generalize the idea to account for unreliable class-attribute associations. Finally, we extend the solution to the "few-shot" setting, where a small number of category-labeled images are also available for training.

We demonstrate the idea on three large datasets of object and scene categories, and show its clear advantages over status quo models. Our results suggest the valuable role attributes can play for low-cost visual category learning, in spite of the inherent difficulty in learning them reliably.

## 2 Related Work

Most existing zero-shot models take a two-stage classification approach: given a novel image, first its attributes are predicted, then its class label is predicted as a function of those attributes. For example, in [3, 18, 30], each unseen object class is described by a binary indicator vector ("signature") over its attributes; a new image is mapped to the unseen class with the signature most similar to its attribute predictions. The probabilistic Direct Attribute Prediction (DAP) method [8] takes a similar form, but adds priors for the classes and attributes and computes a MAP prediction of the unseen class label. A topic model variant is explored in [31]. The DAP model has gained traction and is often used in other work [23, 19, 29]. In all of the above methods, as in ours, training an unseen class amounts to specifying its attribute signature. In contrast to our approach, none of the existing methods account for attribute unreliability when learning an unseen category. As we will see in the results, this has a dramatic impact on generalization.

We stress that attribute *unreliability* is distinct from attribute *strength*. The former (our focus) pertains to how reliable the mid-level classifier is, whereas the latter pertains to how strongly an image exhibits an attribute (*e.g.,* as modeled by relative [19] or probabilistic [8] attributes). PAC bounds on the tolerable error for mid-level classifiers are given in [18], but that work does not propose a solution to mitigate the influence of their uncertainty.

While the above two-stage attribute-based formulation is most common, an alternative zero-shot strategy is to exploit external knowledge about class relationships to adapt classifiers to an unseen class. For example, an unseen object's classifier can be estimated by combining the nearest existing classifiers (trained with images) in the ImageNet hierarchy [23, 14], or by combining classifiers based on label co-occurrences [13]. In a similar spirit, label embeddings [1] or feature embeddings [4] can exploit semantic information for zero-shot predictions. Unlike these models, we focus on defining new categories through *language-based description* (with attributes). This has the advantage of giving a human supervisor direct control on the unseen class's definition, even if its attribute signature is unlike that observed in any existing trained model.

Acknowledging that attribute classifiers are often unreliable, recent work abandons purely *semantic* attributes in favor of discovering mid-level features that are both detectable and discriminative for a set of class labels [11, 22, 26, 15, 30, 27, 1]. However, there is no guarantee that the discovered features will align with semantic properties, particularly "nameable" ones. This typically makes them inapplicable to zero-shot learning, since a human supervisor can no longer define the unseen class with concise semantic terms. Nonetheless, one can attempt to assign semantics post-hoc (*e.g.,* [30]). We demonstrate that our method can benefit zero-shot learning with such discovered (pseudo)-attributes as well.

Our idea for handling unreliable attributes in random forests is related to *fractional tuples* for handling missing values in decision trees [21]. In that approach, points with missing values are distributed down the tree in proportion to the observed values in all other data. Similar concepts are explored in [28] to handle features represented as discrete distributions and in [16] to propagate

instances with soft node memberships. Our approach also entails propagating training instances in proportion to uncertainty. However, our zero-shot scenario is distinct, and, accordingly, the training and testing domains differ in important ways. At training time, rather than build a decision tree from labeled data points, we construct each tree using the unseen classes' attribute signatures. Then, at test time, the inputs are attribute classifier predictions. Furthermore, we show how to propagate both signatures and data points through the tree simultaneously, which makes it possible to account for inter-dependencies among the input dimensions and also enables a few-shot extension.

## 3  Approach

Given a vocabulary of $M$ visual attributes, each unseen class $k$ is described in terms of its attribute signature $A_k$, which is an $M$-dimensional vector where $A_k(i)$ gives the association of attribute $i$ with class $k$.[2] Typically the association values would be binary—meaning that the attribute is always present/absent in the class—but they may also be real-valued when such fine-grained data is available. We model each unseen class with a single signature (*e.g.,* whales are *big* and *gray*). However, it is straightforward to handle the case where a class has a multi-modal definition (*e.g.,* whales are *big* and *gray* OR whales are *big* and *black*), by learning a zero-shot model per "mode". Whether the attribute vocabulary is hand-designed [8, 3, 19, 29, 23] or discovered [30, 11, 22], our approach assumes it is expressive enough to discriminate between the categories.

Suppose there are $K$ unseen classes of interest, for which we have no training images. Our zero-shot method takes as input the $K$ attribute signatures and a dataset of images labeled with attributes, and produces a classifier for each unseen class as output. At test time, the goal is to predict which unseen class appears in a novel image.

In the following, we first describe the initial stage of building the attribute classifiers (Sec. 3.1). Then we introduce a zero-shot random forest trained with attribute signatures (Sec. 3.2). Next we explain how to augment that training procedure to account for attribute unreliability (Sec. 3.2.2) and signature uncertainty (Sec. 3.2.3). Finally, we present an extension to few-shot learning (Sec. 3.3).

### 3.1  Learning the attribute vocabulary

As in any attribute-based zero-shot method [3, 8, 18, 23, 19, 7, 29], we first must train classifiers to predict the presence or absence of each of the $M$ attributes in novel images. Importantly, the images used to train the attribute classifiers may come from a variety of objects/scenes and need not contain any instances of the unseen categories. The fact that attributes are *shared* across category boundaries is precisely what allows zero-shot learning.

We train one SVM per attribute, using a training set of images $\boldsymbol{x}_i$ (represented with standard descriptors) with binary $M$-dimensional label vectors $\boldsymbol{y}_i$, where $\boldsymbol{y}_i(m) = 1$ indicates that attribute $m$ is present in $\boldsymbol{x}_i$. Let $\hat{a}_m(\boldsymbol{x})$ denote the Platt probability score from the $m$-th such SVM applied to test input $\boldsymbol{x}$.

### 3.2  Zero-shot random forests

Next we introduce our key contribution: a random forest model for zero-shot learning.

#### 3.2.1  Basic formulation: Signature random forest

First we define a basic random forest training algorithm for the zero-shot setting. The main idea is to train an ensemble of decision trees using attribute *signatures*—not image descriptors or vectors of attribute predictions. In the zero-shot setting, this is all the training information available. Later, at test time, we *will* have an image in hand, and we will apply the trained random forest to estimate its class posteriors.

Recall that the $k$-th unseen class is defined by its attribute signature $A_k \in \Re^M$. We treat each such signature as the lone positive "exemplar" for its class, and discriminatively train random forests to distinguish all the signatures, $A_1, \ldots, A_K$. We take a one-versus-all approach, training one forest for each unseen class. So, when training class $k$, the $K - 1$ other class signatures are the negatives.

For each class, we build an ensemble of decision trees in a breadth-first manner. Each tree is learned by recursively splitting the signatures into subsets at each node, starting at the root. Let $I_n$ denote an indicator vector of length $K$ that records which signatures appear at node $n$. For the root node, all $K$ signatures are present, so we have $I_n = [1, \dots, 1]$. Following the typical random forest protocol [2], the training instances are recursively split according to a randomized test; it compares one dimension of the signature against a threshold $t$, then propagates each one to the left child $l$ or right child $r$ depending on the outcome, yielding indicator vectors $I_l$ and $I_r$. Specifically, if $I_n(k) = 1$, then if $A_k(m) > t$, we have $I_r(k) = 1$. Otherwise, $I_r(k) = 0$. Further, $I_l = I_n - I_r$.

Thus, during training we must choose two things at each node: the query attribute $m$ and the threshold $t$, represented jointly as the split $(m, t)$. We sample a limited number of $(m, t)$ combinations[3] and choose the one that maximizes the expected information gain $IG_{basic}$:

$$IG_{basic}(m, t) = H(p_{I_n}) - \left( P(A_i(m) > t | I_n(i) = 1) \, H(p_{I_l}) + P(A_i(m) \le t | I_n(i) = 1) \, H(p_{I_r}) \right) \quad (1)$$

$$= H(p_{I_n}) - \left( \frac{\|I_l\|_1}{\|I_n\|_1} H(p_{I_l}) + \frac{\|I_r\|_1}{\|I_n\|_1} H(p_{I_r}) \right), \quad (2)$$

where $H(p) = -\sum_i p(i) \log_2 p(i)$ is the entropy of a distribution $p$. The 1-norm on an indicator vector $I$ sums up the occurrences $I(k)$ of each signature, which for now are binary, $I(k) \in \{0, 1\}$. Since we are training a zero-shot forest to discriminate class $k$ from the rest, the distribution over class labels at node $n$ is a length-2 vector:

$$p_{I_n} = \left[ \frac{I_n(k)}{\|I_n\|_1}, \frac{\sum_{i \ne k} I_n(i)}{\|I_n\|_1} \right]. \quad (3)$$

We grow each tree in the forest to a fixed, maximum depth, terminating a branch prematurely if less than 5% of training samples have reached a node on it. We learn $J = 100$ trees per forest.

Given a novel test image $x_{test}$, we compute its *predicted* attribute signature $\hat{a}(x_{test}) = [\hat{a}_1(x_{test}), \dots, \hat{a}_M(x_{test})]$ by applying the attribute SVMs. Then, to predict the posterior for class $k$, we use $\hat{a}(x_{test})$ to traverse to a leaf node in each tree of $k$'s forest. Let $P_k^j(\ell)$ denote the fraction of positive training instances at a leaf node $\ell$ in tree $j$ of the forest for class $k$. Then $P(k | \hat{a}(x_{test})) = \frac{1}{J} \sum_j P_k^j(\ell)$, the average of the posteriors across the ensemble.

If we somehow had perfect attribute classifiers, this basic zero-shot random forest (in fact, one such tree alone) would be sufficient. Next, we show how to adapt the training procedure defined so far to account for their unreliability.

### 3.2.2 Accounting for attribute prediction unreliability

While our training "exemplars" are the true attribute signatures for each unseen class, the test images will have only approximate estimates of the attributes they contain. We therefore augment the zero-shot random forest to account for this unreliability *during training*. The main idea is to generalize the recursive splitting procedure above such that a given signature can pursue multiple paths down the tree. Critically, those paths will be determined by the false positive/true positive rates of the individual attribute predictors. In this way, we expand each idealized training signature into a distribution in the predicted attribute space. Essentially, this preemptively builds in the appropriate "cushion" of expected errors when choosing discriminative splits.

Implementing this idea requires two primary extensions to the formulation in Sec. 3.2.1: (i) we inject attribute validation data and its associated attribute classification error statistics into the tree formation process, and (ii) we redefine the information gain to account for the partial propagation of training signatures. We explain each of these components in turn next.

First, in addition to signatures, at each node we maintain a set of validation data in order to gauge the error tendencies of each attribute classifier. For the experiments in this paper (Sec 4), our method reserves some attribute classifier training data for this purpose. Denote this set of attribute-labeled images as $\mathcal{D}_V$. During random forest training, this data is recursively propagated down the tree following each split once it is chosen. Let $\mathcal{D}_V(n) \subseteq \mathcal{D}_V$ denote the set of validation data inherited at node $n$. At the root, $\mathcal{D}_V(n) = \mathcal{D}_V$.

With validation data thus injected, we can estimate the test-time receiver operating characteristic (ROC)[4] for an attribute classifier at any node in the tree. For example, the estimated false positive rate at node $n$ for attribute $m$ at threshold $t$ is $\text{FP}(n, m, t) = P_n(\hat{a}_m(\boldsymbol{x}) > t \mid \boldsymbol{y}(m) = 0)$, which is the fraction of examples in $\mathcal{D}_V(n)$ for which the attribute $m$ is absent, but the SVM predicts it to be present at threshold $t$. Here, $\boldsymbol{y}(m)$ denotes the $m$-th attribute's label for image $\boldsymbol{x}$.

For any node $n$, let $I'_n$ be a real-valued indicator vector, such that $I'_n(k) \in [0, 1]$ records the *fractional* occurrence of the training signature for class $k$ at node $n$. At the root node, $I'_n(k) = 1$, $\forall k$. For a split $(m, t)$ at node $n$, a signature $A_k$ splits into the right and left child nodes according to its ROC for attribute $m$ at the operating point specified by $t$. In particular, we have:

$$I'_r(k) = I'_n(k)P_n(\hat{a}_m(\boldsymbol{x}) > t \mid \boldsymbol{y}(m) = A_k(m)), \ \text{ and } \ I'_l(k) = I'_n(k)P_n(\hat{a}_m(\boldsymbol{x}) \leq t \mid \boldsymbol{y}(m) = A_k(m)),$$
(4)

where $\boldsymbol{x} \in \mathcal{D}_V(n)$ . When $A_k(m) = 1$, the probability terms are $\text{TP}(n, m, t)$ and $\text{FN}(n, m, t)$ respectively; when $A_k(m) = 0$, they are $\text{FP}(n, m, t)$ and $\text{TN}(n, m, t)$. In this way, we channel all *predicted* negatives to the left child node. In contrast, a naive random forest (RF) trained on signatures assumes ideal attribute classifiers and channels all *ground truth* negatives—*i.e.,* true negatives *and* false positives—through the left node.

To illustrate the meaning of this fractional propagation, consider a class "elephant" known to have the attribute "gray". If the "gray" attribute classifier fires only on 60% of the "gray" samples in the validation set, *i.e.,* TP=0.6, then only $0.6$ fraction of the "elephant" signature is passed on to the positive (*i.e.,* right) node. This process repeats through more levels until fractions of the single "elephant" signature have reached all leaf nodes. Thus, a single class signature emulates the estimated statistics of a full training set of class-labeled instances with attribute predictions.

We stress two things about the validation data propagation. First, the data in $\mathcal{D}_V$ is labeled by attributes only; *it has no unseen class labels and never features in the information gain computation*. Its only role is to estimate the ROC values. Second, the recursive sub-selection of the validation data is important to capture the dependency of TP/FP rates at higher level splits. For example, if we were to select split $(m, t)$ at the root, then the fractional signatures pushed to the left child must all have $A(m) < t$, meaning that for a candidate split $(m, s)$ at the left child, where $s > t$, the correct TP and FP rates are both 0. This is accounted for when we use $\mathcal{D}_V(n)$ to compute the ROC, but would not have been, had we just used $\mathcal{D}_V$. Thus, our formulation properly accounts for dependencies between attributes when selecting discriminative thresholds, an issue not addressed by existing methods for missing [21] or probabilistically distributed features [28].

Next, we redefine the information gain. When building a zero-shot tree conscious of attribute unreliability, we choose the split maximizing the expected information gain according to the *fractionally* propagated signatures (compare to Eqn. (2)):

$$IG_{zero}(m, t) = H(p_{I'_n}) - \left( \frac{\|I'_l\|_1}{\|I'_n\|_1} H(p_{I'_l}) + \frac{\|I'_r\|_1}{\|I'_n\|_1} H(p_{I'_r}) \right).$$
(5)

The distribution $p_{I'_z}, z \in \{l, r\}$ is computed as in Eqn. (3). For full pseudocode and a schematic illustration of our method, please see supp.

The discriminative splits under this criterion will be those that not only distinguish the unseen classes but also persevere (at test time) as a strong signal in spite of the attribute classifiers' error tendencies. This means the trees will prefer both reliable attributes that are discriminative among the classes, as well as less reliable attributes coupled with intelligently selected operating points that remain distinctive. Furthermore, they will omit splits that, though highly discriminative in terms of idealized signatures, were found to be "unlearnable" among the validation data. For example, in the extreme case, if an attribute classifier cannot distinguish positives and negatives, meaning that TPR=FPR, then the signatures of all classes are equally likely to propagate to the left or right, *i.e.,* $I'_r(k)/I'_n(k) = I'_r(j)/I'_n(j)$ and $I'_l(k)/I'_n(k) = I'_l(j)/I'_n(j)$ for all $k, j$, which yields an information gain of 0 in Eqn. (5) (see supp). Thus, our method, while explicitly making the best of imperfect attribute classification, inherently prefers more learnable attributes.

The proposed approach produces unseen category classifiers with zero category-labeled images. The attribute-labeled validation data is important to our solution's robustness. If that data perfectly represented the true attribute errors on images from the unseen classes (which we cannot access, of course, because images from those classes appear only at test time), then our training procedure would be equivalent to building a random forest on the test samples' attribute classifier outputs.

### 3.2.3 Accounting for class signature uncertainty

Beyond attribute classifier unreliability, our framework can also deal with another source of zero-shot uncertainty: instances of a class often deviate from class-level attribute signatures. To tackle this, we redefine the soft indicators $I'_r$ and $I'_l$ in Eqn. 4, appending a term to account for annotation noise. Please see supp. for details.

### 3.3 Extending to few-shot random forests

Our approach also admits a natural extension to few-shot training. Extensions of zero-shot models to the few-shot setting have been attempted before [31, 26, 14, 1]. In this case, we are given not only attribute signatures, but also a dataset $\mathcal{D}_T$ consisting of a small number of images with their class labels. We essentially use the signatures $A_1, \ldots, A_K$ as a prior for selecting good tree splits that also satisfy the traditional training examples. The information gain on the signatures is as defined in Sec. 3.2.2, while the information gain on the training images, for which we can compute classifier outputs, uses the standard measure defined in Sec. 3.2.1. Using some notation shortcuts, for few-shot training we recursively select the split that maximizes the combined information gain:

$$IG_{few}(m,t) = \lambda\, IG_{zero}(m,t)\{A_1, \ldots, A_K\} + (1-\lambda)\, IG_{basic}(m,t)\{\mathcal{D}_T\}, \qquad (6)$$

where $\lambda$ controls the role of the signature-based prior. Intuitively, we can expect lower values of $\lambda$ to suffice as the size of $\mathcal{D}_T$ increases, since with more training examples we can more precisely learn the class's appearance. This few-shot extension can be interpreted as a new way to learn random forests with descriptive priors.

## 4 Experiments

**Datasets and setup** We use three datasets: (1) Animals with Attributes (**AwA**) [8] ($M = 85$ attributes, $K = 10$ unseen classes, 30,475 total images), (2) aPascal/aYahoo objects (**aPY**) [3] ($M = 65$, $K = 12$, 15,339 images) (3) SUN scene attributes (**SUN**) [20] ($M = 102$, $K = 10$, 14,340 images). These datasets capture a wide array of categories (animals, indoor and outdoor scenes, household objects, etc.) and attributes (parts, affordances, habitats, shapes, materials, etc.). The attribute-labeled images originate from 40, 20, and 707 "seen" classes in each dataset, respectively; we use the class labels solely to map to attribute annotations. We use the unseen class splits specified in [9] for AwA and aPY, and randomly select the 10 unseen classes for SUN (see supp.). For all three, we use the features provided with the datasets, which include color histograms, SIFT, PHOG, and others (see [9, 3, 20] for details).

Following [8], we train attribute SVMs with combined $\chi^2$-kernels, one kernel per feature channel, and set $C = 10$. Our method reserves 20% of the attribute-labeled images as ROC validation data, then pools it with the remaining 80% to train the final attribute classifiers. We stress that our method and all baselines have access to exactly the same amount of attribute-labeled data.

We report results as mean and standard error measured over 20 random trials. Based on cross-validation, we use tree depths of (AwA-9, aPY-6, SUN-8), and generate $(\#m, \#t)$ tests per node (AwA-(10,7), aPY-(8,2), SUN-(4,5)). When too few validation points ($< 10$ positives or negatives) reach a node $n$, we revert to computing statistics over the full validation set $\mathcal{D}_V$ rather than $\mathcal{D}_V(n)$.

**Baselines** In addition to several state-of-the-art published results and ablated variants of our method, we also compare to two baselines: (1) SIGNATURE RF: random forests trained on class-attribute signatures as described in Sec. 3.2.1, without an attribute uncertainty model, and (2) DAP: Direct Attribute Prediction [8, 9], which is a leading attribute-based zero-shot object recognition method widely used in the literature [8, 3, 18, 30, 8, 23, 19, 29].[5]

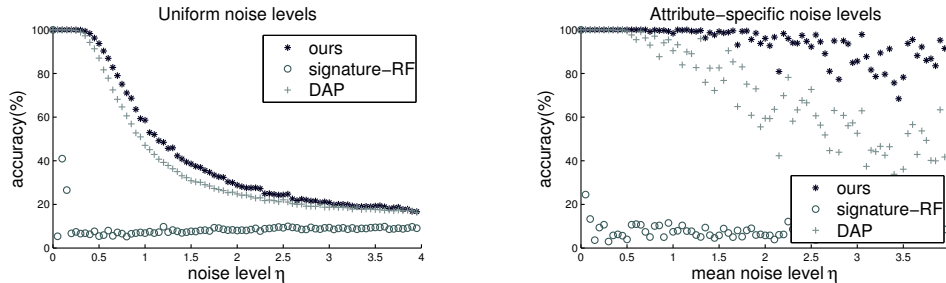

Figure 1: Zero-shot accuracy on AwA as a function of attribute uncertainty, in controlled noise scenarios.

| Method/Dataset | AwA | aPY | SUN |
|---|---|---|---|
| DAP | 40.50 | 18.12 | 52.50 |
| SIGNATURE-RF | $36.65 \pm 0.16$ | $12.70 \pm 0.38$ | $13.20 \pm 0.34$ |
| OURS W/O ROC PROP, SIG UNCERTAINTY | $39.97 \pm 0.09$ | $24.25 \pm 0.18$ | $47.46 \pm 0.29$ |
| OURS W/O SIG UNCERTAINTY | $41.88 \pm 0.08$ | $24.79 \pm 0.11$ | $\mathbf{56.18 \pm 0.27}$ |
| OURS | $\mathbf{43.01 \pm 0.07}$ | $\mathbf{26.02 \pm 0.05}$ | $\mathbf{56.18 \pm 0.27}$ |
| OURS+TRUE ROC | $54.22 \pm 0.03$ | $33.54 \pm 0.07$ | $66.65 \pm 0.31$ |

Table 1: Zero-shot learning accuracy on all three datasets. Accuracy is percentage of correct category predictions on unseen class images, $\pm$ standard error.

## 4.1 Zero-shot object and scene recognition

**Controlled noise experiments** Our approach is designed to overcome the unreliability of attribute classifiers. To glean insight into how it works, we first test it with controlled noise in the test images' attribute predictions. We start with hypothetical perfect attribute classifier scores $\hat{a}_m(\boldsymbol{x}) = A_k(m)$ for $\boldsymbol{x}$ in class $k$, then progressively add noise to represent increasing errors in the predictions. We examine two scenarios: (1) where all attribute classifiers are equally noisy, and (2) where the average noise level varies per attribute. See supp. for details on the noise model.

Figure 1 shows the results using AwA. By definition, all methods are perfectly accurate with zero noise. Once the attributes are unreliable (*i.e.,* noise $> 0$), however, our approach is consistently better. Furthermore, our gains are notably larger in the second scenario where noise levels vary per attribute (right plot), illustrating how our approach properly favors more learnable attributes as discussed in Sec. 3.2.2. In contrast, SIGNATURE-RF is liable to break down with even minor imperfections in attribute prediction. These results affirm that our method benefits from both (1) estimating and accounting for classifier noisiness and (2) avoiding uninformative attribute classifiers.

**Real unreliable attributes experiments** Next we present the key zero-shot results for our method applied to three challenging datasets using over 250 real attribute classifiers. Table 1 shows the results. Our method significantly outperforms the existing DAP method [9]. This is an important result: DAP is today the most commonly used model for zero-shot object recognition, whether using this exact DAP formulation [8, 23, 19, 29] or very similar non-probabilistic variants [3, 30]. Note that our approach beats DAP despite the fact we use only 80% of the attribute-labelled images to train attribute classifiers. This indicates that modeling how good/bad the attribute classifiers are is even more important than having better attribute classifiers. Furthermore, this demonstrates that modeling only the *confidence* of an attribute's presence in a test image (which DAP does) is inadequate; our idea to characterize their *error* tendencies *during training* is valuable.

Our substantial improvements over SIGNATURE-RF also confirm it is imperative to model attribute classifier unreliability. Our gains over DAP are especially large on SUN and aPY, which have fewer positive training samples per attribute, leading to less reliable attribute classifiers—exactly where our method is needed most. On AwA too, we outperform DAP on 7 out of 10 categories, with largest gains on "giant panda"(10.2%),"whale seal"(9.4%) and "persian cat"(7.4%), classes that are very different from the train classes. Further, if we repeat the experiment on AwA reducing to 500 randomly chosen images for attribute training, our overall accuracy gain over DAP widens to 8 points ($28.0 \pm 0.9$ vs. 20.42).

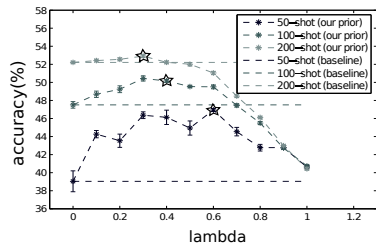

| Method | Accuracy |
|---|---|
| Lampert et al. [8] | 40.5 |
| Yu and Aloimonos [31] | 40.0 |
| Rohrbach et al. [24] | 35.7 |
| Kankuekul et al. [7] | 32.7 |
| Yu et al. [30] | 48.3 |
| OURS (named attributes) | $43.0 \pm 0.07$ |
| OURS (discovered attributes) | $\mathbf{48.7 \pm 0.09}$ |

(a) Few-shot. Stars denote selected $\lambda$.　　　(b) Zero-shot vs. state of the art

Figure 2: (a) Few-shot results. (b) Zero-shot results on AwA compared to the state of the art.

Table 1 also helps isolate the impact of two components of our method: the model of signature uncertainty (see OURS W/O SIG UNCERTAINTY), and the recursive propagation of validation data (see OURS W/O ROC PROP, SIG UNCERTAINTY). For the latter, we further compute TPR/FPRs globally on the full validation dataset $\mathcal{D}_V$ rather than for node-specific subsets $\mathcal{D}_V(n)$. We see both aspects contribute to our full method's best performance (see OURS). Finally, OURS+TRUE ROC provides an "upper bound" on the accuracy achievable with our method for these datasets; this is the result attainable were we to use the unseen class images as validation data $\mathcal{D}_V$. This also points to an interesting direction for future work: to better model expected error rates on images with unseen attribute combinations. Our initial attempts in this regard included focusing validation data on seen class images with signatures most like those of the unseen classes, but the impact was negligible.

Figure 2b compares our method against all published results on AwA, using both named and discovered attributes. When using standard AwA named attributes, our method comfortably outperforms all prior methods. Further, when we use the discovered attributes from [30], it performs comparably to their attribute decoding method, achieving the state-of-the-art on AwA. This result was obtained using a generalization of our method to handle the continuous attribute strength signatures of [30].

### 4.2 Few-shot object and scene recognition

Finally, we demonstrate our few-shot extension. Figure 2a shows the results, as a function of both the amount of labeled training images and the prior-weighting parameter $\lambda$ (cf. Sec 3.3).[6] When $\lambda = 0$, we rely solely on the training images $\mathcal{D}_T$; when $\lambda = 1$, we rely solely on the attribute signatures *i.e.,* zero-shot learning. As a baseline, we compare to a method that uses solely the few training images to learn the unseen classes (dotted lines). We see the clear advantage of our attribute signature prior for few-shot random forest training. Furthermore, we see that, as expected, the optimal $\lambda$ shifts towards 0 as more samples are added. Still, even with 200 training images in $\mathcal{D}_T$, the prior plays a role (e.g., the best $\lambda = 0.3$ on blue curve). The star per curve indicates the $\lambda$ value our method selects automatically with cross-validation.

## 5   Conclusion

We introduced a zero-shot training approach that models unreliable attributes—both due to classifier predictions and uncertainty in their association with unseen classes. Our results on three challenging datasets indicate the method's promise, and suggest that the elegance of zero-shot learning need not be abandoned in spite of the fact that visual attributes remain very difficult to predict reliably. Further, our idea is applicable to other uses of semantic mid-level concepts for higher tasks *e.g.,* poselets for action recognition [12], discriminative mid-level patches for location recognition [27] *etc.*, and in domains outside computer vision. In future work, we plan to develop extensions to accommodate inter-attribute correlations in the random forest tests and multi-label random forests to improve scalability for many unseen classes.

**Acknowledgements:** We thank Christoph Lampert and Felix Yu for helpful discussions and sharing their code. This research is supported in part by NSF IIS-1065390 and ONR ATL.

## Footnotes

[1]and have an attribute vocabulary rich enough to form distinct signatures for each category of interest

[2]We use "class" and "category" to refer to an object or scene, *e.g., zebra* or *beach*, and "attribute" to refer to a property, *e.g., striped* or *sunny*. "Unseen" means we have no training images for that class.

[3]With binary $A_i(m)$, all $0 < t < 1$ are equivalent in Sec 3.2.1. Selecting $t$ becomes important in Sec 3.2.2.

[4]The ROC captures the true positive (TP) vs. false positive (FP) rates (equivalently the true negative (TN) and false negative (FN) rates) as a function of a decision value threshold.

[5]We use the authors' code: http://attributes.kyb.tuebingen.mpg.de/

[6]These are for AwA; see supp. for similar results on the other two datasets.

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
