[Supplementary Material]

# Zero-Shot Recognition with Unreliable Attributes
# (Supplementary material)

Dinesh Jayaraman
dineshj@cs.utexas.edu

Kristen Grauman
grauman@cs.utexas.edu

In this document, we provide supplementary material for our NIPS 2014 paper "Zero-Shot Recognition with Unreliable Attributes".

Sec 1 shows how unlearnable attributes are avoided by our method. Sec 2 discusses the details of the signature uncertainty model introduced in Sec 3.2.3 of the paper. Sec 3 gives additional details for our controlled noise experiments (Sec 4.1 of the paper). Sec 4 lists the 10 SUN database test classes chosen at random. Sec 5 shows more few-shot results, as a continuation of Sec 4.2 in the paper. Sec 6 contains pseudocode for our proposed method, and Sec 7 contains schematics illustrating our method and its ablated variants.

## 1   Unlearnable attributes

As a sanity check, we show how accounting for classifier unreliability as detailed in Sec 3.2.2 of the paper also inherently avoids unlearnable attributes. For the extreme case of completely unlearnable attributes, the classifier cannot tell between positives and negatives, so that TPR=FPR (regardless of threshold). If a candidate split $(m, t)$ tested at any node involves such an attribute $m$, then signatures of all classes are equally likely to propagate to the left or right, i.e., $I'_r(k)/I'_n(k) = I'_r(j)/I'_n(j)$ and $I'_l(k)/I'_n(k) = I'_l(j)/I'_n(j)$ for all $k, j$. In other words, $I'_l$ and $I'_r$ are multiples of $I'_n$. Plugging into Eqn. (3), we see that this means $p_{I'_n} = p_{I'_l} = p_{I'_r}$.

Further plugging this into Eqn. (5), we see:

$$
\begin{aligned}
IG_{zero} &= H(p_{I'_n}) - \left( \frac{\|I'_l\|_1}{\|I'_n\|_1} H(p_{I'_l}) + \frac{\|I'_r\|_1}{\|I'_n\|_1} H(p_{I'_r}) \right) & (1) \\
&= H(p_{I'_n}) - \left( \frac{\|I'_l\|_1}{\|I'_n\|_1} H(p_{I'_n}) + \frac{\|I'_r\|_1}{\|I'_n\|_1} H(p_{I'_n}) \right) & (2) \\
&= H(p_{I'_n}) - \left( \frac{\|I'_l\|_1}{\|I'_n\|_1} H(p_{I'_n}) + \left( 1 - \frac{\|I'_l\|_1}{\|I'_n\|_1} \right) H(p_{I'_n}) \right) & (3) \\
&= H(p_{I'_n}) - H(p_{I'_n}) = 0. & (4)
\end{aligned}
$$

Since $IG_{zero}$ is constrained to be $\geq 0$, this split will never be chosen by our method.

## 2   Class signature uncertainty

In Sec 3.2.3 of the paper, we summarize a method to deal with uncertainty in class-attribute signatures. This is achieved by appropriately modifying the soft indicator vectors, which in implementation terms, amounts to adding perturbed copies of each "exemplar" signature to the training set. We now describe the former in detail, and show how it is equivalent to the latter.

Repeating Eqn 4 from the paper, when we assume perfect class signatures, we set:

$$I'_l(k) = I'_n(k)P(\hat{a}_m(\boldsymbol{x}) > t \mid \boldsymbol{y}(m) = A_k(m)), \text{ and } I'_r(k) = I'_n(k)P(\hat{a}_m(\boldsymbol{x}) \leq t \mid \boldsymbol{y}(m) = A_k(m)), \quad (5)$$

where the probabilities are simply the TPR and FNR respectively on the validation data subset $\mathcal{D}_V(n)$ at node $n$ respectively. Now, to account for the class signature uncertainty, we expand out the probabilities in terms of the TPR/FNR and a new term reflecting the signature uncertainty. Specifically, denote by $a(\boldsymbol{x})$ the *true* attribute

signature of instance $\boldsymbol{x} \in \mathcal{D}_V(n)$ as opposed to its annotated signature $\boldsymbol{y}$. Then,

$$P(\hat{a}_m(\boldsymbol{x}) > t \mid \boldsymbol{y}(m) = A_k(m)) \quad = \quad \sum_s P(\hat{a}_m(\boldsymbol{x}) > t \mid a_m(\boldsymbol{x}) = s) \, P(a_m(\boldsymbol{x}) = s \mid \boldsymbol{y}(m) = A_k(m)) \tag{6}$$

and

$$P(\hat{a}_m(\boldsymbol{x}) \le t \mid \boldsymbol{y}(m) = A_k(m)) \quad = \quad \sum_s P(\hat{a}_m(\boldsymbol{x}) \le t \mid a_m(\boldsymbol{x}) = s) \, P(a_m(\boldsymbol{x}) = s \mid \boldsymbol{y}(m) = A_k(m)) \tag{7}$$

where $s$ runs over all possible values of the $a_m(\boldsymbol{x})$. The first terms on the RHS in the above equations represent the familiar TPR and FNR respectively (computed from the validation data), while the second term captures the non-trivial dependency between the true attribute value and the annotation. The above changes in the computation of probabilities exactly model the effect of training data expansion by adding an infinite number of perturbed variants of the attribute signatures, perturbed as per $P(a_m(\boldsymbol{x}) = s \mid \boldsymbol{y}(m) = A_k(m))$.

In expanding the probabilities thus, we have implicitly assumed the following structure for dependencies among $\hat{a}_m(\boldsymbol{x})$, $a_m(\boldsymbol{x})$ and $A_k(m)$ :

$$p(\hat{a}_m(\boldsymbol{x}), a_m(\boldsymbol{x}), A_k(m)) = p(\hat{a}_m(\boldsymbol{x}) \mid a_m(\boldsymbol{x})) \, p(a_m(\boldsymbol{x}) \mid A_k(m)) \, p(A_k(m)) \tag{8}$$

With other dependency assumptions, it is possible to derive variants of this method with differences in the probability expansions of Eqn 6 and 7.

In implementing this attribute uncertainty model, we observed that it was generally very common for instances labeled positive for a given attribute to be actually negative (due to occlusions *etc*) but the reverse was uncommon. This is understandable because we use class-level associations *e.g., images* of class "person" may not "have hands" for any number of reasons, while the class "person" does "have hands" as per its class-level attribute signature. For this reason, we restricted ourselves to flipping (a fraction of) only the positive bits in the attribute signatures. Based on cross-validation we flipped 0.15, 0.3 and 0.0 fractions of the bits on AwA, aPY and SUN respectively. On SUN alone, zero-shot recognition does not benefit from modeling uncertainty in attribute annotations. We believe that this is because there is little scope for in-class variation in attribute signatures among the SUN scenes, since the attributes are of four types: "functional affordances", "materials", "surface properties" and "spatial envelope" [1]. All these types of attributes are closely related to the scene labels themselves, and are unlikely to be missing from class instances for reasons such as occlusion, since they are not usually localized to specific parts of instance images *e.g.,* images belonging to the "mountain" category (from SUN) are nearly always marked with the affordance "climbing"(attribute in SUN) - it is very unlikely that the "climbing" affordance would be taken away because of the way a mountain is pictured. In contrast, say a "person" (category in aPY) may have images without "hands" (attribute in aPY) as discussed above, simply because of occlusions.

## 3   Noise model

The synthetic attribute classifier scores used in Sec 4.1 of the paper from the paper are constructed by corrupting a hypothetical perfect attribute classifier's scores with progressively increasing noise. Specifically, for noise setting 0, our synthetic attribute classifier scores are $\hat{a}_m(\boldsymbol{x}) = A_k(m)$, for $\boldsymbol{x}$ in class $k$. For noise level $\eta$, we (1) decrease scores on positive samples and (2) increase scores on negative samples by adding noise as follows: $\hat{a}_m(\boldsymbol{x}) = A_k(m) + (-1)^{A_k(m)}(n \bmod 1)$, where $n$ is drawn from the exponential distribution with mean $\eta$: $p_\eta(n) = \eta^{-1} e^{-n/\eta}$. The $\bmod$ keeps all scores in $[0, 1]$.

For the two scenarios in the paper, shown in Fig 1 of the paper, we did the following. For scenario 1 (equally noisy classifiers), all classifiers were corrupted with noise drawn from an exponential distribution with mean $\eta$ (as above). For scenario 2 (attribute-specific noise levels), we draw the mean noise $\eta'$ of each attribute classifier itself from a new exponential distribution whose mean is $\eta$ (plotted along the x-axis in Fig 1, right side).

## 4   SUN test classes

The ten SUN test classes picked at random were: "inn/indoor","flea market/indoor", "lab classroom", "outhouse/outdoor", "chemical plant", "mineshaft", "lake/natural", "shoe shop", "art school" and "archive".

Figure 1: Few-shot results for (left) aPY and (right) SUN: Overall trends are similar to those obtained for AwA

(a) assuming ideal classifiers    (b) accounting for unreliability    (c) validation data propagation

Figure 2: (Best viewed in color, and zoomed in) The SIGNATURE-RF in (a) treats signatures as exemplar training instances directly. This does not account for attribute prediction errors (false positives and negatives) at test time. (b) shows how this may be corrected with FPR and TPR of attribute classifiers, estimated on validation data. Further, by propagating the validation data down the tree to get node-specific validation sets as in (c), interdependencies in attribute error statistics are accounted for. Proposed changes from the previous panel are highlighted in (b) and (c). For this illustration, we have assumed $K = 4$ unseen classes. Of these, the first two classes do not have attribute $m$ in their class-level attribute signatures, while the last two do, as shown in (a). For (b) and (c), we have assumed $\text{TPR}(n, m, t) = 0.7, \text{FPR}(n, m, t) = 0.4$.

# 5    Few-shot results

Fig 1 shows the few-shot results for aPY and SUN with 50 and 100 shots, similar to Fig 2a in the paper. Interestingly, on SUN, our zero-shot learning approach beats 100-shot attribute prediction based learning (on the 10 test classes). Overall trends remain similar to those on AwA, discussed in the paper.

# 6    Pseudocode

Pseudocode for training a tree in our proposed zero-shot random forest is given in algorithm block 1.

# 7    Illustration of proposed method

A schematic illustration of the proposed zero-shot random forest learning technique is shown in Fig 2. The panels also represent the ablated variants of our method used in Table 1 of the paper.

# References

[1]  G Patterson and J Hays. SUN Attribute Database: Discovering, Annotating, and Recognizing Scene Attributes. In *CVPR*, 2012.

---
**Algorithm 1** Training procedure for a tree in our zero-shot random forest
---
**procedure** TREETRAIN
**Given**:
    $M$ attribute classifiers
    $K$ attribute signatures $A_k$ of length $M$, one for each target class $k$
    Validation data $\mathcal{D}_V$, annotated with ground truth attributes alone
**Notation**:
    $l(n), r(n)$ denote left and right children of node $n$, respectively
    Nodes are numbered in a breadth-first manner *i.e.,* root node 0, first level nodes $1, 2$ *etc*
    $\hat{a}_m(\boldsymbol{x})$ denotes Platt score for input $\boldsymbol{x}$ exhibiting attribute $m$
**Do**:
    $\mathcal{D}_V(0) \leftarrow \mathcal{D}_V,$
    $n \leftarrow 0, I_0(k) = 1, \forall 1 \le k \le K$                 ▷ Initialization
    **while** (not branch termination condition) **do**       ▷ Learn each node in turn
        $IG_{max} \leftarrow 0, m^* \leftarrow 0, t^* \leftarrow 0$
        **for** i=1 to d **do**
            Sample (attribute $m$, threshold $t$)            ▷ Sample candidate split
            $\text{FPR}(n, m, t) \leftarrow \frac{|\{(\boldsymbol{x}^v, \boldsymbol{y}^v) \in \mathcal{D}_V(n) \mid \hat{a}_m(\boldsymbol{x}^v) > t, \boldsymbol{y}^v(m) = 0\}|}{|\{(\boldsymbol{x}^v, \boldsymbol{y}^v) \in \mathcal{D}_V(n) \mid \boldsymbol{y}^v(m) = 0\}|}$ *etc*
            Compute $I_l'$ and $I_r'$ as defined in Eqn. (4) in the paper (to model signature uncertainty too, instead
use the indicator vectors defined in Eqn. (5), (6) and (7) in the supplementary).
            Compute $IG_{zero}(m, t)$ as defined in Eqn. (5) in the paper.       ▷ Evaluate candidate split
            **if** $IG_{zero}(m, t) > IG_{max}$ **then**
                $IG_{max} \leftarrow IG_{zero}, m^* \leftarrow m, t^* \leftarrow t$       ▷ update best split
                $I'_{l(n)} \leftarrow I_l', I'_{r(n)} \leftarrow I_r'$     ▷ update the fractional signatures routed to children nodes
        $\mathcal{D}_V(l(n)) \leftarrow \{(\boldsymbol{x}^v, \boldsymbol{y}^v) \in \mathcal{D}_V(n) \mid \hat{a}_{m^*}(\boldsymbol{x}) \le t^*\}$
        $\mathcal{D}_V(r(n)) \leftarrow \mathcal{D}_V(n) \setminus \mathcal{D}_V(l(n))$
        $n \leftarrow n + 1$                   ▷ move to next node
---