[Reviews · NeurIPS 2014]

Submitted by Assigned_Reviewer_27

The paper strives to bridge the gap between the theory and practice of attribute-based zero-shot learning. The theory is that novel classes can be recognized automatically using pre-trained attribute predictors; in practice, however, learning these attribute classifiers can be as difficult or even more so than learning the object classes themselves.

Random forests are trained to predict unseen classes from attribute vectors,
and the training procedure takes into account the reliability of the attribute
detectors by propagating a validation set through each decision tree at training
time. The authors show how the method can be extended to handle training with a
few training examples of test categories. The method achieves state-of-the-art
results on several attribute datasets.

QUALITY: The authors do a really nice job of handling this fundamental zero-shot learning problem using a random forest framework. The model is elegant and theoretically sound. Results on 3 standard datasets are strong. The authors do a nice job of performing ablation studies, introducing artificial noise, evaluating several setting (zero- versus few-shot learning), and comparing with the literature.

CLARITY: The paper is very well written. One confusion was in Section 3.2.1: when the threshold t is introduced, the attribute signatures are still binary, so t can be any value 0 < t < 1 without changing anything in equation (2). Then it is not clear in lines 207-212 how a novel test example can be propagated down the tree, since t seems to be ill-defined. This is cleared up in later sections.

ORIGINALITY: This is a nice application of a random forest framework to an important problem.

SIGNIFICANCE: The paper addresses a fundamental problem in zero-shot learning.
Summary: The paper is strong, interesting, and sound. The results on 3 datasets in a variety of settings are convincing. The paper deserves to be published.

Submitted by Assigned_Reviewer_33

This paper proposes a new random forest based method for zero-shot learning that better accommodates the uncertainty in attribute classifier predictions. Uncertainty is measured via the performance (true positive rate, false positive rate, etc.,) of attribute classifiers on a held-out validation set. The paper proposes to use this performance information in the information gain computation during the learning of the random forest (for pursuing a given category-attribute signature along multiple paths of a decision tree), thereby accommodating the "uncertainty" of attribute predictions.

Quality: The proposed ideas are supported well with relevant empirical/quantitative analysis. However it lacks in qualitative analysis (For e.g., I would have liked to see the types of attributes and the categories that are effected/improved by this method.) Also it is unclear how many of the parameters involved were selected (for e.g., why was the 80%-20% split for training-validation chosen, etc.,)

Clarity: The paper is well-written and easy to read.

Originality: While most previous works have addressed the problem of "attribute strength", this paper claims to focus on the lesser explored problem of "attribute reliability". Although many learning algorithms/techniques implicitly account for classifier unreliability, I believe this work is novel in terms of explicit unreliability handling.

Significance: While the proposed method (random forest) is not new, the idea (of modeling unreliability by measuring classifier performance on validation data to improve zero-shot learning performance) introduced in this paper is interesting/thought-provoking to researchers working on attributes.
Summary: While the idea and the method introduced in this paper are not significantly novel, their application towards improving zero-shot learning is revealing/interesting.

Submitted by Assigned_Reviewer_34

This paper presents a method for attribute-based zero-shot learning using random forests. Attribute classifiers are first trained using SVMs. A random forest is trained on class-attribute signature vectors, while using a validation set of images with attribute-labels is used to estimate attribute reliability at each node of the tree. An extension to few-shot learning is also proposed.

Clarity: The paper is very clear and well-written. It does a good job summarizing related work.

Originality: The novelty of this paper is somewhat low, but not too low to be accepted. One issue is that the claim that the proposed approach deals with unreliable attribute predictions while earlier work does not is a bit overstated or misleading. For example, direct attribute prediction [7] (DAP)--the baseline method used in this paper and many others--has some similar abilities to deal with attribute uncertainty. Both approaches use the basic approach:
a) Train attribute classifiers on image features (identical for both papers)
b) Train a 2nd class probability estimator as a function of attribute predictions, using a validation set to estimate the reliability of attribute classifiers ([6] uses Platt scaling and a probabilistic model to combine attributes, whereas the proposed approach uses a new random forest-based method)

In this way, I don't find the proposed approach to be a fundamentally new way of handling attribute uncertainty. What it does differently than earlier work is that it uses a more complex model for modelling attribute noise: 1) it uses an ROC-based estimate instead of a sigmoidal model like Platt scaling, and 2) whereas [6] treats attributes as independent, the proposed approach effectively models joint statistics in attribute errors, since the validation set is propagated down trees. In this way, I think it would be a little more appropriate to describe the approach as a better way to model attribute correlation (I don't know this area well, but Scheirer et al. "Multi-Attribute Spaces: Calibration for Attribute Fusion and Similarity Search" might be one related paper that could be cited).

At the same time, one possible weakness of the proposed approach in comparison to methods that treat attributes independently is that it seems like one would need a large validation set to obtain valid probability estimates as it gets subdivided when propagated down decision trees. This issue was glossed over in the paper, and proposed approach for dealing with this (line 323) seems heuristic. It would have been nice to see more experiments evaluating the effect of training and validation set size.

Quality: In general, the quality of the paper is good and the technical choices and descriptions make sense. The experiments are appropriate and support the claims of the paper, although the level of quantitative improvement isn't high enough to accept the paper solely on the merit of empirical results. The extension to few-shot learning (Section 3.3) felt less polished than the rest of the paper. Whereas the rest of the paper was clear and explanations were lengthy, this section was very brief. To me, a method that could combine class-attribute priors and image attribute estimates when estimating probabilities p(a|...) might make more intuitive sense then a weighted combination of information gains (Eq. 6).
Summary: This is a well written paper, with technical decisions that make sense, and adequate experiments; however, the main selling point of the paper as a new way of handling unreliable attributes is somewhat questionable.
Author Feedback
Author rebuttal: Thanks to all reviewers for their valuable comments.

AR27:

* Threshold “t”, Lines 207-212
Right, “t” in 3.2.1 could be fixed to any value without changing anything. We introduce the notation here so that it extends easily to Sec 3.2.2. We will clarify to avoid confusion.

AR33:

* "Quantitative analysis supports ideas well; Qualitatively, what types of attributes/categories are affected most?"
We can elaborate in a final version (and briefly here). For example, we outperform DAP in 7 of 10 AwA categories, with the largest F-score gains on panda and seal (~10%) and the least on pig and leopard (~-1%). We see our gains are highest on unseen classes that are more distant from train classes, i.e., greater domain shift makes it more important to account for attribute unreliability. We also find that simply preferring “reliable” (high AUC) attributes in a random forest (RF) is inferior to not only our method but also the SIGNATURE-RF baseline, yet there *is* strong correlation between attribute usage in our RFs and attribute classifier AUC. These two observations together are evidence that our principled combination of predictability and discriminativeness in IG calculation is critical to accuracy. Finally, while nodes near the tops of trees often use high-AUC attributes, lower nodes select more diverse attributes, since ROC behavior on subsets of data reaching those nodes differs from overall validation ROC.

* "Parameter - How was 80-20 split selected?"
The 80-20 split was chosen to have enough validation data to estimate ROC values without reducing attribute classifier training data significantly (and see point ## below). Parameters to the learning algorithm (#m, #t) are selected with cross-validation (see L320-4).

* “Although many algorithms implicitly account for classifier unreliability, I believe this work is novel in terms of explicit unreliability handling.”
Yes, and our results show that "implicitly accounting for unreliability" merely by having soft attribute assignments (as in DAP [7]) is inferior to our method. As AR33 notes, it is essential to distinguish between attribute *unreliability* (accounted for by our method) and *strength* (accounted for by probabilistic attributes in the DAP model [7] or relative attributes [17]). Please see L86-90.

* “While proposed method (random forest) is not new, the idea … is interesting/thought-provoking…”
To formulate a zero-shot training procedure that accounts for unreliable attributes, we introduce several novel technical components beyond a standard “vanilla” random forest (RF): 1) we define zero-shot RFs that can be trained without actual training data using only one “attribute signature” per class, 2) we develop the first training procedure that addresses *joint* error tendency statistics of attributes 3) our method is the first to explicitly address noise in linguistic descriptions for zero-shot training 4) we define a few-shot extension to accommodate both data instances and external linguistic descriptions.

AR34:

* "Originality: DAP [7] has similar abilities to deal with attribute uncertainty."
It is important to distinguish attribute *strength* from attribute prediction *unreliability*. DAP handles only the former, we handle the latter. See L86-90. Yes, DAP [7] uses probabilistic predictions of attributes, but nowhere in its learning procedure does it model the *error tendencies* of those classifiers. In fact, our method also uses the same probabilistic attribute predictions (from code by [7]), then builds on top of them to account for their TPR/FPRs, thus capturing unreliability of the strength predictions. Our results on 3 datasets show the clear impact (see Tab 1, DAP vs. OURS): by representing likely errors of attribute predictions during training, we obtain more robust zero-shot recognition results.

* "Main selling point as new way to handle unreliable attributes is somewhat questionable"
Again, we emphasize that the prior zero-shot learning work does not handle unreliable attributes. Rather, the prior work represents only the attribute “strengths”. Our very idea to represent attribute error tendencies when building the object category classifiers is novel.

* "Novelty of proposed method: 1) more complex model of attribute noise (ROC instead of Platt) and 2) models joint statistics in attribute errors"
Yes, these are important distinctions, as explained in L79-90, 120-2, 218-224, 249-257. We can further emphasize in discussion.

* "Scheirer et al. might be related"
That work is concerned with calibrating probabilistic attribute *strength predictions* (e.g. [7,17]) to compare them better for image search, and is only tenuously linked to our work. Yes, we too address joint statistics, but of *error tendencies* in attribute predictions and with the goal of predicting unseen object classes in zero-shot learning.

* "May need large validation set"
Using validation sets of about 1400-4800 labeled images (20% of data), our method provides superior results compared to all the baselines and existing methods on 3 datasets. We stress that all methods have access to exactly the same attribute-labeled data -- our method never uses any additional data, it just reserves some to form its validation set (L316-9).

## * "Would be nice to evaluate effect of validation set size"
While we selected an 80-20 split heuristically in our submission, we find accuracy is quite robust to this choice. On AwA, results for various train-validation splits are: 50-50 (42.06), 60-40 (42.16), 70-30 (42.71), 80-20 (43.01), 90-10 (43.16). Note how performance is stable with less validation data, and margins of our gains vs baselines persist with just 10% of data used for validation.

* "Few-shot: could try to combine class-attribute priors"
Our current few-shot formulation is promising (see Fig 2a & Supp), but certainly other variations could make for interesting future work. Thanks for the suggestion.